# Positive and Negative Aspects of Sodium Intake in Dialysis and Non-Dialysis CKD Patients

**DOI:** 10.3390/nu13030951

**Published:** 2021-03-16

**Authors:** Yasuyuki Nagasawa

**Affiliations:** Department of Internal Medicine, Division of Kidney and Dialysis, Hyogo College of Medicine, 1-1 Mukogawa-Cho, Nishinomiya, Hyogo 663-8501, Japan; nagasawa@hyo-med.ac.jp; Tel.: +81-798-45-6521; Fax: +81-798-45-6880

**Keywords:** hypertension, body weight, mortality, sodium, dialysis

## Abstract

Sodium intake theoretically has dual effects on both non-dialysis chronic kidney disease (CKD) patients and dialysis patients. One negatively affects mortality by increasing proteinuria and blood pressure. The other positively affects mortality by ameliorating nutritional status through appetite induced by salt intake and the amount of food itself, which is proportional to the amount of salt under the same salty taste. Sodium restriction with enough water intake easily causes hyponatremia in CKD and dialysis patients. Moreover, the balance of these dual effects in dialysis patients is likely different from their balance in non-dialysis CKD patients because dialysis patients lose kidney function. Sodium intake is strongly related to water intake via the thirst center. Therefore, sodium intake is strongly related to extracellular fluid volume, blood pressure, appetite, nutritional status, and mortality. To decrease mortality in both non-dialysis and dialysis CKD patients, sodium restriction is an essential and important factor that can be changed by the patients themselves. However, under sodium restriction, it is important to maintain the balance of negative and positive effects from sodium intake not only in dialysis and non-dialysis CKD patients but also in the general population.

## 1. Introduction

Sodium intake is related to extracellular fluid volume. Increased fluid volume increases blood pressure, resulting in hypertension. Moreover, excretion of sodium is also related to blood pressure. Increased sodium excretion in the kidneys requires elevated blood pressure in the glomeruli, also resulting in hypertension. Guyton described this phenomenon as a pressure–natriuresis relationship [1]. Therefore, sodium intake is strongly related to fluid volume (body weight) and hypertension. Hypertension is an important key factor in determining mortality. On the other hand, sodium intake is also related to nutrition. If someone eats more food, that person is usually ingesting more sodium. Moreover, sodium intake is correlated with appetite. Therefore, sodium intake is strongly related to nutrition, which is another important key factor determining mortality. There are two contradictory aspects of the effect of sodium intake: hypertension and nutrition (see Figure 1). This review describes the interaction between sodium intake, body weight, hypertension, nutrition, and mortality in chronic kidney disease patients, including dialysis patients.

## 2. Interaction between Sodium Intake and Body Weight (Extracellular Volume) in Non-Dialysis CKD Patients

In extracellular fluid volume, osmotic presser is derived from sodium, glucose, and blood urea nitrogen concentrations. Among these osmolytes, sodium is the strongest determinate of osmotic pressure because sodium occupies a major portion of osmolytes in extracellular volume. After sodium intake, osmotic pressure should increase, resulting in strong thirst via the thirst center in the brain (see Figure 2). Water intake continues until osmotic pressure became normal, similar to physiological saline. After drinking water, extracellular volume increases to the same osmotic pressure as normal osmotic pressure in the extracellular fluid. After ingesting 9 g of salt, a person needs to drink 1000 mL of water to reach 0.9% physiological saline in the extracellular fluid (This person is ideal and fictional for this review). Edema is often observed after intake of excess sodium. In the general population, excretion of sodium in the kidneys increases just after consuming sodium [1]. Sodium excretion rapidly equalizes to sodium intake. However, in non-dialysis chronic kidney disease (CKD) patients, decreased kidney function prolongs the effect of sodium intake because the ability to excrete excess sodium is decreased, resulting in elevated blood pressure.

## 3. Interaction between Sodium Intake and Hypertension in Non-Dialysis CKD Patients

After sodium intake, the kidney increases sodium excretion via inducing elevated blood pressure. Known as Guyton’s pressure–natriuresis curve [1], salt excretion is basically proportional to blood pressure. From renal aspects, blood pressure depends upon arterial resistance to glomeruli, such as renal artery stenosis, the coefficient of glomerular excess filtration and sodium reabsorption in distal tubules, regulated primarily by aldosterone (see Figure 3). The coefficient of glomerular excess filtration depends on the effective glomerular filtration surface and coefficient of penetration. In CKD patients, the effective glomerular filtration surface decreases, resulting in enhanced salt sensation. Indeed, CKD patients are reported to have strong salt sensation in response to food [2,3,4,5]. In the setting of increased salt sensitivity, excess sodium intake easily elevates blood pressure. Excess salt intake increases proteinuria via elevated internal glomerular pressure. Hypertension and proteinuria are important factors in determining the rate of worsening renal function. Decreasing blood pressure and proteinuria are important therapeutic targets. Excess sodium intake worsens both blood pressure and proteinuria, and inhibition of sodium intake decreases blood pressure and proteinuria at the same time (see Figure 4). Restriction of sodium intake is an important therapy in CKD patients.

## 4. Interaction among Sodium Intake, Appetite, and Nutrition in Non-Dialysis CKD Patients

The sensation of taste components consists of five factors: salt, sweet, sour, bitter, and umami. Salt is an important taste component and influences other components. Salt restriction also causes taste restriction. Moreover, salt and bitter taste acuity declines with age, while sweet and sour perception does not [6]. The prevalence of CKD in elderly patients is much higher than in the young population [7,8]. Therefore, most CKD patients suffer from low salt sensation. Moreover, CKD itself worsens the accuracy of salt and sour taste sensations [9,10], while the mechanism this abnormality of tastes induced by CKD remained unclear. Weakening salt sensation theoretically leads to more salt intake during free food intake. Typically, salt restriction worsens taste, and therefore causes loss of appetite. In CKD patients, this effect becomes larger due to decreased salt sensation. Under the same salt concentrations, salt restriction also means diet restriction. Salt intake in CKD patients is strongly correlated with phosphate intake, which is equal to protein intake [11]. Sodium restriction easily causes appetite loss and worsens nutrition status. However, salt restriction improves salt sensation [10,12]. Kusaba-K et al. reported that one week of sodium restriction dramatically improved salt sensation. Several reports have also supported that education on sodium intake improves salt sensation itself [9,10]. It was reported that educational hospitalization with enhanced salt reduction guidance reduced the risk of end-stage renal disease [13]. Sodium restriction seems to be effective under the control of nutritional status.

## 5. Interaction between Sodium Intake and Mortality in Non-Dialysis CKD Patients

Generally, excess sodium intake is considered an important life-threatening risk factor that can be improved by personal efforts. Excess salt intake necessitates induction of high blood pressure to increase salt excretion in the kidney according to the pressure-diuresis curve [1]. Excess sodium is believed to increase blood pressure, resulting in worsening mortality. According to this theory, the World Health Organization recommended restrictions on salt intake for the general population [14,15]. However, Martin O’Donnell et al. reported a U-curve effect of sodium intake on morality in the general population [16] (see Figure 5). This article reported that among 103,570 international subjects in eighteen countries, the relationship between sodium and potassium intake was evaluated by morning fasting urine with respect to mortality during a median follow-up of 8.2 years. Among the joint sodium and potassium excretion categories, the lowest risk of death and cardiovascular events occurred in the group with moderate sodium excretion and higher potassium excretion (3–5 g sodium/day = 7.6–12.7 g salt/day; 21.9% of cohort). Compared to this reference group, the combinations of low potassium with low sodium excretion (<3 g sodium/day ≤ 7.6 g salt/day) (hazard ratio 1.23 (1.11–1.37); 7.4% of cohort) and low potassium with high sodium excretion (>5 g sodium/day ≥ 12.7 g salt/day) (1.21 (1.11–1.32); 13.8% of cohort) were associated with the highest risk, followed by low sodium excretion and higher potassium excretion (<3 g sodium/day ≤ 7.6 g salt/day) (1.19 (1.02–1.38); 3.3% of cohort), and high sodium excretion and higher potassium excretion (1.10 (1.02–1.180); 29.6% of cohort). Higher potassium excretion attenuated the increased cardiovascular risk associated with high sodium excretion (P for interaction = 0.007). This report concluded that moderate sodium intake combined with high potassium intake is associated with the lowest risk of mortality and cardiovascular events. This J-shaped effect of sodium intake on mortality has also been reported in patients with cardiovascular events [17]. Martin O’Donnell et al. reported this phenomenon in the general population in several studies [18,19,20]. The finding that potassium intake attenuated the effect of sodium intake suggested that the low food intake induced by sodium restriction worsened mortality. However, after adjustment for nutritional factors, the effect of low sodium intake was still observed. The effect of sodium restriction, except for nutritional factors, ultimately remains unknown. However, one considerable mechanism is that imbalance of sodium intake and water intake may cause hyponatremia. Water intake is usually encouraged to avoid dehydration [21], especially in elderly subjects. Excess water intake, along with restricted sodium intake requires water diuresis with salt reabsorption in the kidney. In patients with several diseases, such as heart failure, chronic kidney disease, and liver failure, the kidney cannot provide sufficient water diuresis because high vasopressin levels induced by these disease conditions perturb proper water diuresis, resulting in hyponatremia. Hyponatremia is well known as a risk factor for worse mortality [22,23,24,25,26]. Of course, overload of sodium intake worsens mortality not only in heart failure patients but also in the general populations. Moreover, the effect of an overload of excess sodium intake in heart failure patients was markedly higher than in the general population [17]. Sodium restriction is essential in patients with many kinds of diseases, but in the case of sodium restriction, regular monitoring of serum sodium concentrations is also essential. Based on this evidence, sodium restriction in the general population has become controversial [27].

In non-dialysis CKD patients, the effect of excess sodium intake is more important than in the general population. Excess sodium intake increases blood pressure, especially glomerular blood pressure. Elevated permeable pressure in the glomeruli induced by excess sodium intake increases proteinuria [28], which is an important and classical risk factor for kidney disease progression. Moreover, excess sodium intake suppresses the renin–angiotensin system, resulting in cancelling the reno-protective effect of renin–angiotensin system blockers. Renin–angiotensin system blockers in conjunction with sodium restriction can decrease proteinuria more effectively than renin–angiotensin system blockers alone [28,29] (see Figure 4). Even in the case of diuretic use, which may decrease the amount of sodium in the body, sodium restriction could decrease proteinuria [28]. Theoretically, sodium restriction in chronic kidney disease patients should decrease proteinuria and blood pressure and therefore improve renal progression, resulting in improved mortality. The CRIC study, comprising an important chronic kidney disease cohort, reported the relationship between sodium intake evaluated by urinary sodium excretion and kidney disease progression or mortality [30]. Non-dialysis CKD patients with high sodium intake (more than 194.6 mmol sodium/day = 11.4 g salt/day) exhibited significantly worse renal prognosis. However, CKD patients with several categories of sodium intakes less than 194.6 mmol sodium/day (11.4 g salt/day) had almost the same kidney prognosis. Moreover, those CKD patients had the same mortality (see Figure 5). Apparently, non-dialysis CKD patients were more vulnerable to high salt intake than the general population (see Figure 5), but there were no significant differences between CKD patients with low salt intake and those with moderate salt intake, most likely because CKD patients are a high-risk group of hyponatremia who were encouraged to drink water to avoid dehydration, and sodium restriction may easily cause hyponatremia, resulting in a worsening prognosis. In CKD patients, mild sodium restriction (less than 194.6 mmol = 11.4 salt/day) is essential with regular sodium concentration monitoring.

## 6. Summary of Interaction between Sodium Intake, Body Weight, Hypertension, Nutrition, and Mortality in Non-Dialysis CKD Patients

Sodium intake theoretically has dual effects on prognosis, including both harmful effects and beneficial effects (see Figure 1). Sodium intake is believed to induce hypertension, resulting in poor prognosis. In contrast, sodium intake increases appetite, resulting in improved nutrition and good prognosis. In fact, a U-shaped relationship between sodium intake and mortality support these dual effects in the general population (see Figure 5).

In CKD patients, sodium intake exerts more important effects on prognosis. Sodium intake increases not only blood pressure but also proteinuria, resulting in both poor renal prognosis and life expectancy. In contrast, sodium intake increases appetite, resulting in good nutritional status. In dialysis patients, the obesity paradox is well known [31,32]. This paradox involves a higher body mass index (BMI) causing a better prognosis in dialysis patients, while in the general population, a higher BMI causes a poor prognosis by increasing cardiovascular events [33]. This obesity paradox is also observed in non-dialysis CKD patients [34] (see Figure 6). Another Japanese non-dialysis CKD cohort also reported that low BMI (18.4–20.3) was associated with significant risk of all-cause mortality and infection-related death [35]. Therefore, nutritional factors are very important for CKD patients. Obesity is considered to have negative effects on the progression of kidney disease through elevation of eGFR, hypertension, and proteinuria. Exercise has reno-protective effects [36,37]. However, obesity seemed to cancel out these renoprotective effects [38]. It is important to maintain a balance between nutritionally good conditions and bad conditions induced by obesity. Sodium restriction is necessary for CKD patients to decrease proteinuria and preserve kidney function, but in cases of sodium restriction, nutritional factors, and hyponatremia should be considered.

## 7. Interaction between Sodium Intake and Body Weight (Intradialysis Increase in Body Weight) in Dialysis CKD Patients

In dialysis patients, salt and water intake should be balanced (see Figure 2). When dialysis patients ingest salt for additional taste during meals, the sodium concentration increases. In such cases, sodium concentrations tend to increase and cause strong thirst via the thirst center in the brain. This thirst continues until the sodium concentration becomes normal, similar to physiological saline solution (isotonic sodium chloride solution). Sodium intake should be completely balanced with water intake via the thirst center. Nine grams of sodium chloride intake requires subsequent intake of 1000 mL water, resulting in physiological saline (0.9% sodium chloride solution). In non-dialysis CKD patients, urine attenuates this volume increase induced by sodium, but in dialysis patients, sodium intake induces exactly this volume retention because the kidneys are not functioning properly.

In dialysis patients, salt removal should also be balanced with water removal (see In dialysis patients, salt removal should also be balanced with water removal (see Figure 7). The sodium concentration in dialysate is usually the same as the serum sodium concentration (140 mEq/L), meaning that during dialysis, sodium cannot move to the dialysate. During dialysis, water is usually removed. This water includes sodium, whose concentration should be equal to the serum sodium concentration. Water and salt are removed at the same time during the dialysis process as physiological saline is removed. Therefore, in dialysis patients the intake of sodium and water is basically equal to the removal of sodium and water during the dialysis session because these patients have lost kidney function (see Figure 2 and Figure 7).

Generally, estimation of sodium intake is very difficult because the precise intake and sodium concentration of each meal are usually unknown. However, estimation of water intake, including water in food, is far easier because intradialysis body weight gain is equal to the water intake. The amount of salt intake can be estimated using water intake because salt intake is basically balanced with water intake. Reports of the Japanese dialysis patient registry, which are provided by the Japanese Society of Dialysis Therapy, published the relationship between the increase in body weight during the dialysis interval and mortality one year later. A 3% body weight increase to a 7% body weight increase reportedly resulted in a good prognosis, according to the relationship between body weight increase and mortality after adjustment for fundamental factors, the amount of dialysis therapy and nutritional factors (see Figure 8). If the body weight of the dialysis patient was 60 kg, a 3% body weight increase meant 1800 g. Therefore, 1800 g divided by 3 days equals to 600 g, and 600 g divided by 1000 mL physiological saline, including 9 g salt, yields 5.4 g salt. This calculation means that approximately 6 g salt (=2.36 g sodium) per day is essential for a 60 kg dialysis patient. In contrast, in 60 kg dialysis patients, a 7% body weight increase was 4200 g. 4200 g divided by 3 days equals 1400 g, and 1400 g divided by 1000 mL physiological saline, including 9 g sodium chloride, yields 12.6 g salt. This calculation means that 60 kg dialysis patients can ingest 12.6 g salt (=4.96 g sodium) per day safely. (These 60 kg dialysis patients are ideal and fictional for this review).

There is a pitfall in the method to determine the recommended salt intake. The reason the standard salt intake can be calculated by intradialysis body weight gain depends on thirst. If people drink water when they are not thirsty, the estimation of sodium intake based on intradialysis body weight gain would not make sense. Drinking without thirst typically involves alcohol drinks. If a patient drinks a 350 g can of beer every day, these patients drank a total of 1050 g of water during the interval of dialysis (3 days), resulting in a total salt intake of 9.4 g during those periods. If a patient drank two bottles of beer every day, these patients ingested 3798 g of water during hemodialysis intervals, resulting in a salt intake of 11.3 g a day, which means that this patient could not eat anything with salt, except for the beer. (These drunken dialysis patients are ideal and fictional for this review). If the patient truly wants to drink alcohol, the patient should choose an alcohol drink including a high concentration of alcohol, such as whisky without water. If dialysis patients drink a small amount of alcohol, their thirst after eating food with salt should be suppressed, and their serum sodium concentration with stay within normal ranges. In the case of sick individuals, patients tend to ingest food, including much water, such as soup, rice porridge (Okayu), and oatmeal. These kinds of food also reduce the capacity of salt intake.

In the case of hypernatremia, strong thirst induces water consumption, resulting in normalization of hypernatremia. This system can work in both normal subjects and hemodialysis patients. In contrast, in the case of hyponatremia, activation of the renin–angiotensin–aldosterone system along with inhibition of vasopressin induces upregulation of salt retention in the kidney without water retention, resulting in normalization of hyponatremia. This system works in subjects without kidney disease, but in dialysis patients, this system does not work because patients have lost kidney function. Moreover, salt removal in dialysis patients is performed by salt with the removal of water during dialysis therapy. The balance between the removal of salt and water is basically constant to ensure serum sodium concentrations similar to physiological saline. Under hyponatremic conditions, thirst cannot correct the balance between water intake and salt intake. Therefore, dialysis patients are prone to developing hyponatremia. The Japan Society of Dialysis Therapy reported that predialysis hyponatremia was associated with worse mortality after one year (see Figure 9). Moreover, hyponatremia in dialysis patients can be corrected by dialysis therapy because sodium moves from the dialysate to serum according to the sodium concentration gradient. This means that dialysis patients with hyponatremia receive salt through dialysis therapy, while those patients restrict salt in food. Rapid correction of serum sodium by dialysis therapy increases serum osmolality, possibly resulting in several ill reactions, such as general fatigue, headache, osmotic demyelination syndrome, and so on. Hyponatremia itself has been reported to have a strong relationship with worse prognosis in several disease conditions [22,23,24,25,26]. If hyponatremia is observed in dialysis patients, the balance between water intake and salt intake should be checked. Overdose of water intake or overrestriction of salt intake along with relative overdose of water results in hyponatremia.

## 8. Interaction between Sodium Intake and Hypertension in Dialysis CKD Patients

Sodium intake is directly related to extracellular fluid volume in dialysis patients via the thirst center, as shown in Figure 2. Increased fluid volume caused the elevation of blood pressure. Indeed, blood pressure just before dialysis session tended to be higher than after dialysis session. Elasticity of blood vessels can buffer this volume effect, but atherosclerosis, which is very common in dialysis patents, prevents this buffering effect. Therefore, hypertension before dialysis session is very common, but in terms of this hypertension in dialysis patients, there is another paradox regarding hypertension in the general population [39,40]. A reverse effect of blood pressure on mortality in dialysis patients was reported in 56,388 incident dialysis patients [41]. In this report, dialysis patients whose blood pressure was less than 120 mmHg had the worst prognosis, those with 120–140 mmHg blood pressure had the second worst prognosis, and there were no differences in prognosis in those with 140–160, 160–180, 180–200, and more than 200 mmHg blood pressure. Another study also reported that in 37,069 dialysis patients, those with less than 115 mmHg systolic blood pressure had the worst mortality, while those with more than 135 mmHg systolic pressure had a good prognosis [42]. Another study in 5433 dialysis patients reported that systolic pressure greater than 180 mmHg improved prognosis [43]. While hypertension itself has been classically considered a risk factor for cardiovascular events [44,45,46,47,48], the role of hypertension as a risk factor for mortality has been controversial [43,49,50,51,52,53] The reverse effect of blood pressure on mortality in dialysis patients is similar to the reverse effect of BMI on mortality, which is well known as the obesity paradox and is discussed above. One of the possible mechanisms is that after initiation of dialysis, nutritional factors, which can be related to BMI and hypertension induced by intradialysis food and sodium intake, are more important than classical cardiovascular event risk factors [54]. Another possible mechanism is that hypertension before dialysis session makes dialysis therapy safe and comfortable until end of scheduled dialysis session time because hypotension during dialysis session sometimes required the termination of dialysis session. KDOQI comments on the 2017 ACC/AHA hypertension guidelines described that the treatment of targets of hypertension in dialysis patients cannot be proposed due to the lack of clinical trial evidence [55]. The Japanese Society of Hypertension described that the Japanese guidelines for hypertension cannot be applied to dialysis patients due to the specific and distinct conditions of dialysis patients.

## 9. Interaction among Sodium Intake, Appetite and Nutrition in Dialysis CKD Patients

Salt sensation in dialysis patients is reported to be worse than that in non-dialysis CKD patients [56]. Theoretically, dialysis patients with salt taste dysfunction ingest more salt than is appropriate to gain sufficient salty taste, possibly resulting in an increase in intradialytic body weight gain. However, there were no changes in intradialytic body weight gain between dialysis patients with or without salt sensation dysfunction [57]. This report indicates that food intake, including salt intake, depends not only on salt sensation dysfunction but also on other factors. One possible factor related to appetite is zinc. Zinc deficiency is common in dialysis patients, and zinc deficiency causes a decrease in the sensation of taste, including salt taste. Moreover, zinc is related to many enzymatic activities related to protein synthesis; therefore, zinc deficiency is reportedly related to nutritional status. In fact, zinc supplementation ameliorated nutritional status [58] in a randomized control study. If dialysis patients suffer from appetite loss and a small increase in intradialysis body weight gain, there is the possibility that too much salt restriction retards food intake through appetite loss, which may also be caused by zinc deficiency.

## 10. Interaction between Sodium Intake and Body Weight (Nutritional Status) and Mortality in Dialysis CKD Patients

Estimation of the amount of sodium intake is difficult; therefore, reports on the relationship between sodium intake in the diet and mortality are limited. Post hoc analysis of the HEMO study reported that increased dietary sodium intake, which was estimated using a history of food intake by dietitians, was independently associated with greater mortality among hemodialysis subjects [59]. Although there were no descriptions of sodium intake in quartiles, only the fourth category of dialysis patients who might ingest more than 4 g sodium a day (=10 g salt intake a day) exhibited significantly higher risk. Moreover, this article reported that sodium intake was strongly associated with caloric intake, but the adjustment of nutritional status was not sufficient. In this study, only weight was reported rather than BMI, while albumin was selected.

Sodium intake is strongly correlated with food intake. Under the same salt sensation, sodium intake is proportional to food intake. Sufficient food intake ameliorates nutritional status. The question is whether a better nutritional status can overcome the negative effects caused by excess sodium intake, such as an increase in intradialytic body weight and hypertension. As well-known as the obesity paradox, high BMI in dialysis patients attenuates mortality (see Figure 6). This obesity paradox was first reported in heart failure patients [60]. Recently, the obesity paradox was observed in many energy wasting diseases, such as COPD and cancers, especially esophageal cancer [33]. The reason the obesity paradox is observed in many energy-wasting diseases is that a stock of energy can sustain energy-wasting conditions and make it possible to overcome several illness conditions, such as infectious diseases and cardiovascular diseases [54]. In dialysis patients, body weight gain ameliorated mortality. Surprisingly, in obesity dialysis patients whose BMI was greater than 30, body weight gain indicated better prognosis [61], and body weight loss indicated poor prognosis. The improvement of prognosis induced by body weight was attenuated by BMI, but good nutritional status could overcome the negative effects induced by obesity, even in obese dialysis patients with a BMI greater than 30. Unless excess sodium intake ameliorates the nutritional condition, sodium intake itself should be acceptable.

## 11. Summary of Interactions between Sodium Intake, Body Weight, Hypertension, Nutrition, and Mortality in Dialysis CKD Patients

In dialysis CKD patients, sodium intake is directly related to intradialytic body weight gain via the thirst center, resulting in hypertension before hemodialysis sessions. However, this hypertension does not worsen the prognosis of dialysis patients. Sodium intake is also directly related to nutrition status via the appetite center. Moreover, the amount of sodium intake is basically proportional to food intake under the same salt taste conditions. Therefore, sodium intake may improve nutritional status. In dialysis patients, both obesity and body weight gain improve prognosis through good nutritional conditions. Sodium intake has both good and bad effects in dialysis CKD patients, as well as in non-CKD patients. However, in dialysis patients, the good effect of sodium basically overcomes the bad effect from the perspective of mortality.

A comfortable dialysis session and safe control of life-limiting biochemical parameters may exert opposite effects from the good nutritional status induced by food intake. Salt intake causes an increase in intradialytic body weight gain, requiring prolonged dialysis time because the time-averaged removal of water is limited. Protein intake, including phosphate intake, may cause hyperphosphoremia, increasing the risk of cardiovascular events. Fruit and vegetable intake may cause hyperkalemia [62], which is famously a risk factor for sudden death (see Figure 10). The standard nutritional intake has not been changed for a long time (see Table 1). This standard is similar to the nutritional guidelines in many countries [63,64,65]. The reason for the unchanged nutritional guidelines was that these guidelines were based on comfortable and safe control of dialysis session and the lives of dialysis patients, which were physiologically unchanged. However, recent advances in drug- and technology-related dialysis therapy may change the physiological limitation of nutritional intake. New phosphate binders and calcium-sensing receptor agonists have become available, which increase protein intake [66,67]. Indeed, protein intake with phosphate binders could improve nutritional factors in the short term [68]. New potassium binders have also become available, making safe fruit and vegetable intake possible [69]. Moreover, it was recently reported that a high frequency of fruit and vegetable intake in dialysis patients improved prognosis [70]. Fruits and vegetables are considered to have good effects on mortality [71]. Potassium intake was not related to hyperkalemia in either non-dialysis CKD patients or dialysis patients [72]. The upper limit of sodium is much higher than the target of sodium intake in the nutritional standard, as discussed in the above section. Food intake, including sodium intake, should be encouraged in dialysis CKD patients to improve nutritional status, resulting in a better prognosis.

## 12. Conclusions: Important Gaps in Awareness of Sodium Intake between Non-Dialysis and Dialysis CKD Patients

Sodium intake has both a good effect on improving nutritional effects and a bad effect on increasing fluid volume, which causes hypertension and increases proteinuria. In non-dialysis CKD patients, doctors should pay attention to the balance between the positive and negative effects of sodium restriction, while in dialysis patients, the good effect of sodium intake basically overcomes the bad effect of sodium intake because nutritional status is an important issue in dialysis patients.

Recently, frailty and protein-energy wasting have been recognized as serious problems in dialysis patients [73,74]. Frailty worsens prognosis in dialysis and non-dialysis CKD patients [75]. Nutritional intervention in frail dialysis patients is an important and practical therapy [76,77]. However, restriction of sodium intake sometimes prevents sufficient food intake. In dialysis CKD patients, a good effect of sodium intake basically overcomes the bad effect because nutritional status limits the prognosis of dialysis patients, while in non-CKD patients, the balance between the positive and negative effects of sodium intake is important because restriction of sodium decreases blood pressure and proteinuria, resulting in preserving kidney function. There was no evidence of the beginning of the change in sodium intake between non-dialysis CKD patients and dialysis patients. However, the report that weight loss in non-dialysis patients worsened prognosis after initiation of dialysis therapy [78] indicated that sodium restriction in non-dialysis CKD patients should sometimes be re-evaluated and relaxed at the time of weight loss as well as during hyponatremia. There are important gaps in awareness of sodium intake between non-dialysis and dialysis CKD patients. Sodium restriction may be allowed in cases of maintaining good nutritional status and normal serum sodium levels in CKD patients, including dialysis patients.

## Figures and Tables

**Figure 1 nutrients-13-00951-f001:**
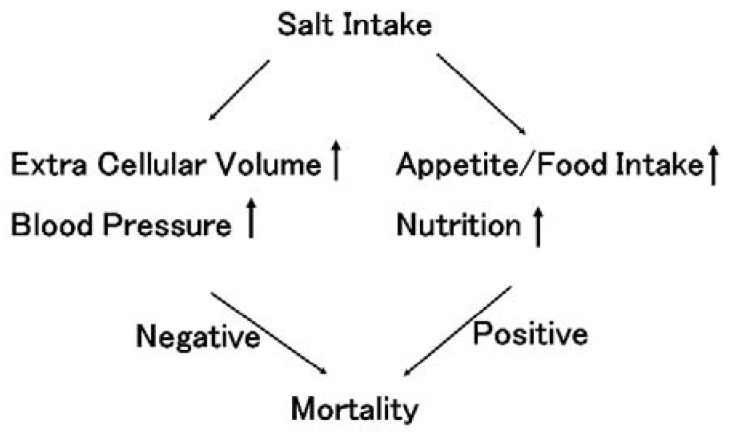
The influence of salt intake on mortality. Salt intake increases extracellular volume, resulting in increased blood pressure in both the general population and chronic kidney disease (CKD) patients, including dialysis patients. Hypertension is considered to worsen prognosis. This salt sensation higher in CKD patients than in the general population. In contrast, salt intake increase appetite. Under the same salt taste, salt intake is basically proportional to the amount of food intake. Good nutritional conditions improve prognosis. Therefore, salt intake has dual effects on prognosis, both negative and positive effects.

**Figure 2 nutrients-13-00951-f002:**
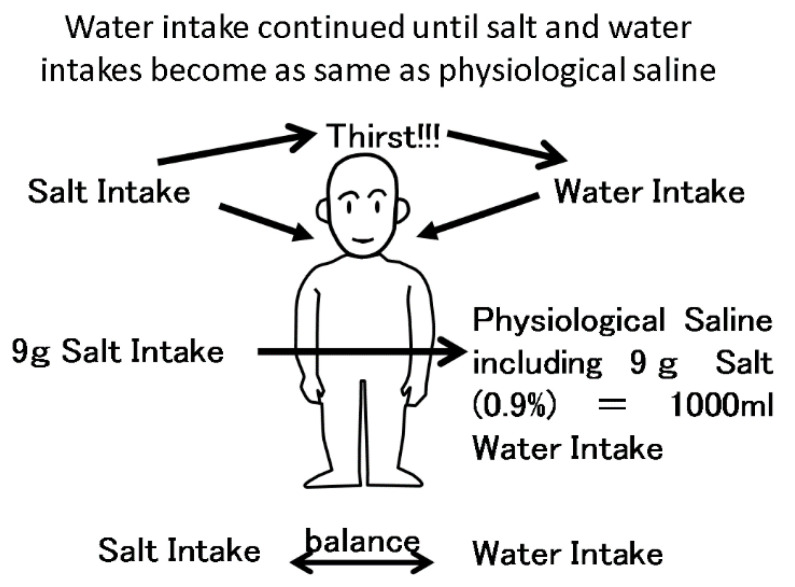
The balance between salt intake and water intake. Salt intake is balanced with water intake via the thirst center. After 9 g of salt intake, a person wants to drink 1000 mL water, resulting in 0.9% physiological saline in extracellular fluid. If the person does not drink enough water, the concentration of sodium will increase, resulting in strong thirst.

**Figure 3 nutrients-13-00951-f003:**
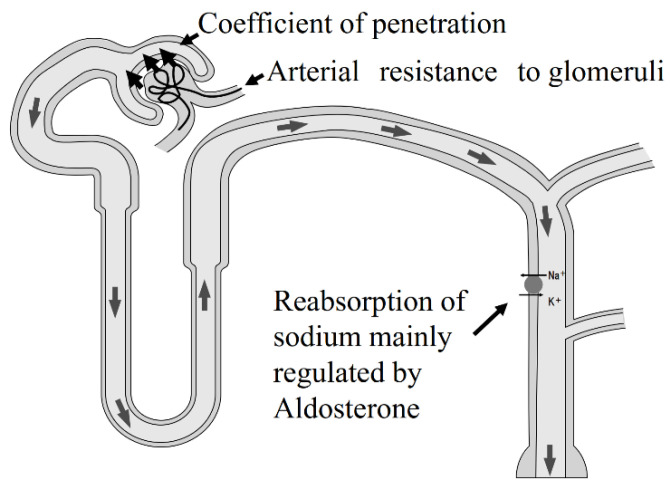
Determinates of blood pressure in the glomeruli. There are three elements that can determine blood pressure. One is arterial resistance to glomeruli, such as renovascular hypertension. The second is the coefficient of penetration, such as hypertension, in CKD patients. The third element is reabsorption of sodium in distal tubules, regulated primarily by aldosterone, such as primary aldosteronism. In CKD patients, the coefficient of penetration is not normal because this coefficient depends on the effective surface of the glomeruli.

**Figure 4 nutrients-13-00951-f004:**
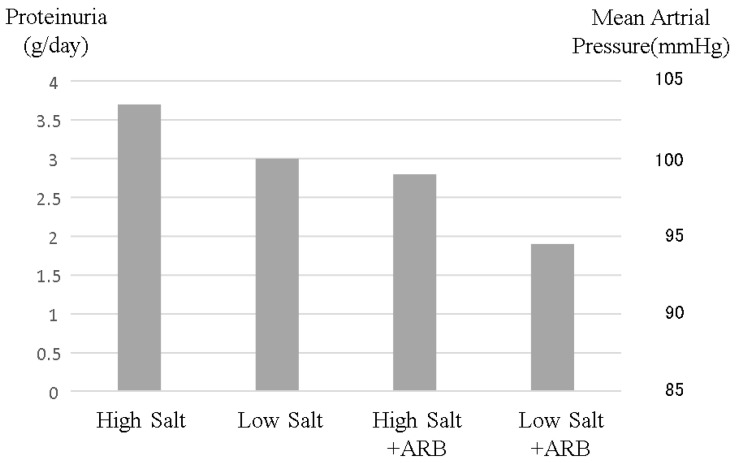
Changes in proteinuria and blood pressure induced by salt intake and angiotensin-receptor blocker administration. High salt intake increases both blood pressure and proteinuria. Although angiotensin-receptor blockers are well known to reduce blood pressure and proteinuria, resulting in reno-protective effects, high salt intake reduces these renoprotective effects. This image was made from data reported by Vogt et al. [28], but this phenomenon is common in CKD patients.

**Figure 5 nutrients-13-00951-f005:**
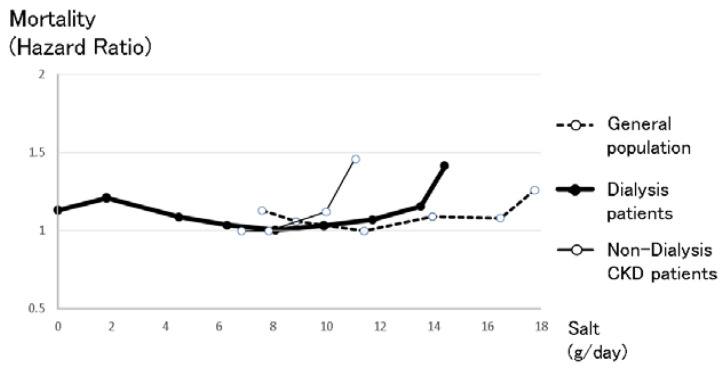
The relationship between salt intake and mortality in the general population, non-dialysis CKD patients and dialysis patients. The right line shows the relationship between salt intake and mortality in the general population, which was made from data reported by O’Donnell et al. [16]. Intake of more than 17.8 g salt/day made the prognosis significantly worse after diet adjustments, while intake of less than 7.6 g salt/day also made the prognosis worse. The center bold line shows the relationship between salt intake and mortality in dialysis patients, which was made from the relationship between intradialysis body weight gain and mortality reported by the Japan Society of Dialysis Therapy. Intake of more than 12.6 g salt/day made the prognosis significantly worse after diet adjustments, while intake of less than 5.4 g salt/day also made the prognosis worse. The left line shows the relationship between salt intake and mortality in non-dialysis CKD patients, which was made from data reported by the CRIC cohort study [30]. Non-dialysis CKD patients were more vulnerable to salt intake than the other groups, likely because excess sodium intake increases blood pressure and proteinuria, resulting in worsening prognosis. Simple comparisons should be considered because these three lines refer to different cohorts that originate in different countries, with different observation periods and numbers.

**Figure 6 nutrients-13-00951-f006:**
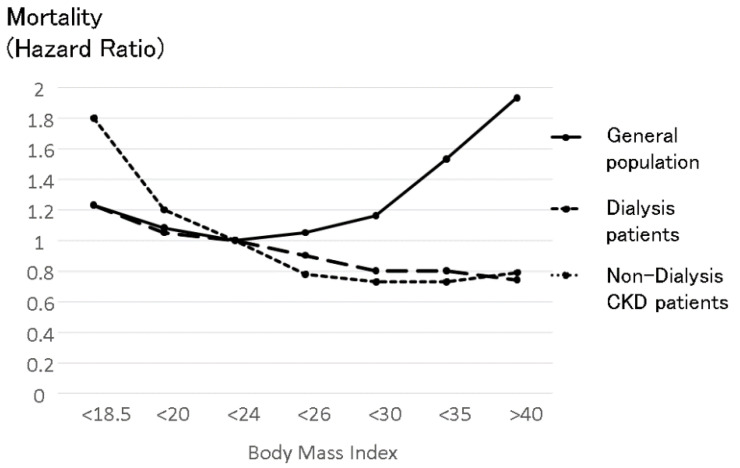
The relationship between body mass index and mortality in the general population, non-dialysis CKD patients and dialysis patients. The upper U-shaped line shows the relationship between body mass index (BMI) and mortality in the general population, which was made from data reported based on 900,000 subjects [33]. The line indicates the average mortality in men and in women. The effect of BMI on mortality in women was smaller than in men in all BMI categories. High BMI was associated with high mortality in the general population, while low BMI was also associated with high mortality. The middle line shows the relationship between BMI and mortality in dialysis patients reported by Kalantar-Zadeh et al. [31]. Higher BMI in dialysis patients consistently improves prognosis in dialysis patients. This divergence from the relationship in the general population was named the “obesity paradox”. The bottom line shows the relationship between BMI and mortality in non-dialysis CKD patients reported by Navaneethan-SD et al. [34]. The obesity paradox was observed in non-dialysis patients. Simple comparisons should be considered because these three lines refer to different cohorts that have different countries of origin, observation periods, and numbers.

**Figure 7 nutrients-13-00951-f007:**
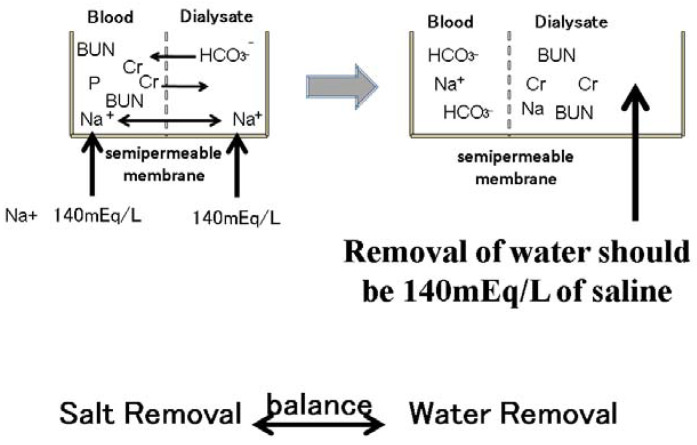
Removal of sodium during dialysis sessions. Small molecular substances, such as blood urea nitrogen and phosphate creatinine, are moved to the dialysate according to the concentration gradient. Bicarbonate also moves to the blood from the dialysate according to the concentration gradient. The sodium concentration in the dialysate is 140 mEq/L. The serum sodium concentration is also approximately 140 mEq/L. Therefore, sodium cannot move from the blood to the dialysate through the semipermeable membrane in the dialyzer. However, water removal during dialysis sessions includes 140 mEq/L sodium. Water removal during dialysis sessions means sodium removal at the same time.

**Figure 8 nutrients-13-00951-f008:**
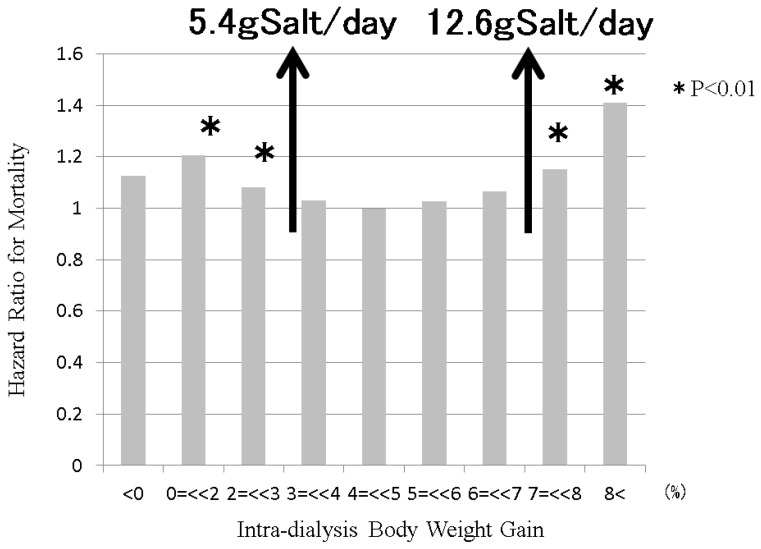
The relationship between intradialysis body weight gain and mortality in dialysis patients. Sodium intake is balanced with water intake via the thirst center. Sodium removal is balanced with water removal during dialysis session. Therefore, intradialysis salt intake can be estimated from intradialysis body weight gain. The Japanese Society of Dialysis Therapy reported the relationship between intradialysis body weight gain and mortality at 1 year as surveillance of dialysis patients by the Japan Society of Dialysis Therapy 2009.12.31. From 3% to 7% of intradialysis body weight gain, the mortality of dialysis patients remained good. From estimations from this intradialysis body weight gain, dialysis patients can maintain good conditions from 5.4 g salt/day (=2.1 g sodium/day) intake to 12.6 g salt/day (=5.0 g sodium/day) intake, if the body weight of the dialysis patient is 60 kg.

**Figure 9 nutrients-13-00951-f009:**
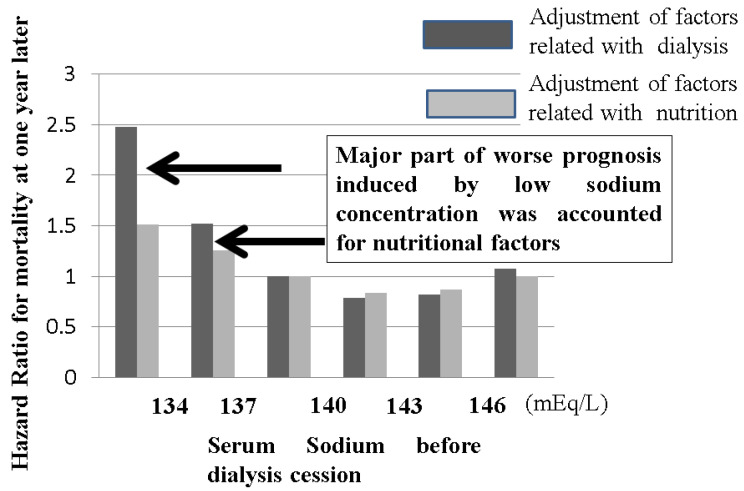
The relationship between serum sodium concentration before dialysis sessions and mortality in dialysis patients. The Japanese Society of Dialysis Therapy reported the relationship between serum sodium concentration before dialysis sessions and mortality at 1 year as surveillance of dialysis patients by the Japan Society of Dialysis Therapy 2009.12.31. Dialysis patients with less than 137 mEq/L sodium concentrations exhibited poor prognosis, but this negative effect induced by low sodium concentration could be adjusted by nutritional factors, meaning that the low sodium concentration in dialysis patients usually indicates poor nutritional intake. Water intake without proper salt easily induces hyponatremia in dialysis patients. In the case of hyponatremia in dialysis patients, the balance between sodium and water intake should be monitored to improve nutritional status.

**Figure 10 nutrients-13-00951-f010:**
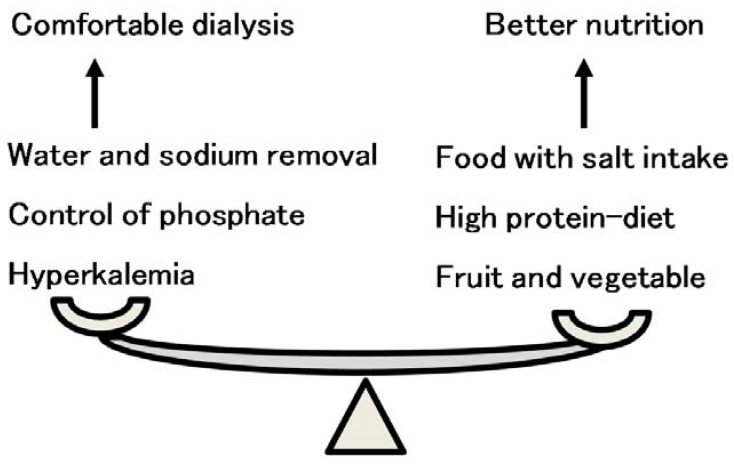
The balance of food and salt intake between comfortable dialysis and better nutrition. Comfortable dialysis therapy is sometimes contrary to better nutritional intake. Sufficient food intake with proper salt taste increases intradialytic body weight gain, resulting in prolonged dialysis session time. A high-protein diet easily induces hyperphosphatemia, resulting in vascular calcification and cardiovascular events. Healthy fruit and vegetable intake induces hyperkalemia.

**Table 1 nutrients-13-00951-t001:** Standard nutritional intake in hemodialysis patients in Japan.

Energy	30–35 kcal/kg ^(*)(**)^
Protein	0.9–1.2 g/kg ^(*)^
Salt	less than 6 g ^(***)^
Water	Minimum requirement
Potassium	less than 2000 mg
Phosphate	less than Protein (g) × 15 mg

* Standard Body Weight (BMI = 22) ** Depending on sex, age, physical activity *** Depending on urine volume, physical activity, body weight, nutritional status, increase of body weight between HD sessions. This recommendation was published by a committee for nutritional factors in the Japanese Society of Dialysis Therapy, including the authors. The original recommendation was written by Nakao, T., Kanno, Y., Nagasawa, Y., Kanazawa, Y., Akib, T., Sanaka, K., Standard nutritional intake in maintained dialysis patients. *J. Jpn. Soc. Dial. Ther.*
**2014**, *47*, 287–291. (In Japanese) [65].

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
