# Peer review of "Positive and Negative Aspects of Sodium Intake in Dialysis and Non-Dialysis CKD Patients"

_nutrients, 2021, doi:10.3390/nu13030951_

Round 1
Reviewer 1 Report
General comments:
While the content of the figures is good, I think that some of the figures need editing for aesthetic improvements. I would consider using a bioillustration software of some type to improve the figures.
In many cases, there seems to be some repetitive statements throughout the manuscript. Please consider reorganizing in some way to avoid this.
The word “cession” is used instead of “session” several times.
In many instances throughout the manuscript, I believe a reference is needed for some of the information.
Please see the following:
Line 70, Line 94, Line 334 (section on alcohol consumption in dialysis patients)
Line 112: Can you please provide clarification on how CKD lowers salt and sour taste sensations?
Line 115: I would consider changing the word “sensitivity” to “sensation.”
Line 129: I think the work “excess” may be appropriate in front of the word sodium.
Line 136: The article that is discussed in this section has both sodium and potassium intake discussed. Is it possible to include a graph that includes both sodium and potassium intake?
Line 189: the expression “several sodium intakes less than…” wording is confusing. Please consider revising.
Line 447: “more than 4 g sodium intake=10 g sodium intake”. Is the equal supposed to be something else?
Author Response
Dear Editor and reviewers
Thank you for careful reading and your important comments upon this review article. According to your comments, I revised the manuscript. The points-by points replies to the comments were described as below. I hope that this revised manuscript is sufficient for publication.
Yasuyuki Nagasawa.
REVIEW1 While the content of the figures is good, I think that some of the figures need editing for aesthetic improvements. I would consider using a bioillustration software of some type to improve the figures.
Ansewer
Thank you for your valuable comments. I updated the figure 1, 3 as your comments. I added the acknowledgements about the support for these figures as shown below.
I thank Mrs. Noriko Echigoya, Division of Kidney and Dialysis, Department of Internal Medicine, Hyogo College of Medicine, for assistance of illustrations.
In many cases, there seems to be some repetitive statements throughout the manuscript. Please consider reorganizing in some way to avoid this.
Ansewer
Thank you for your important comments. This manuscript wrote the effect of sodium in non-dialysis patients and in dialysis patients. Fundamental effects of sodium are common both in non-dialysis patients and in dialysis patients, although there were important differences of the effect between non-dialysis patients and in dialysis patients. These common effects were described repetitively as you commented. I eliminated following words as shown below.
described by Gyton
according to Gyton’s sodium-diuresis curve
The word “cession” is used instead of “session” several times.
Answer
Thank you for your important comments. I replaced the word in 7 places. These were written in red words in revised high-lighted version.
In many instances throughout the manuscript, I believe a reference is needed for some of the information. Please see the following: Line 70, Line 94, Line 334 (section on alcohol consumption in dialysis patients)
Ansewer
The instances in the manuscript were fictional and ideal cases. For example, person who takes 9g Salt was fictional and ideal case. I added the descriptions as shown below.
(This person is ideal and fictional for this review)
(These 60Kg dialysis patients are ideal and fictional for this review.)
(These drunken dialysis patients are ideal and fictional for this review.)
Line 112: Can you please provide clarification on how CKD lowers salt and sour taste sensations?
Ansewer
The reason why CKD lowered the salt and sour taste was unknown, maybe because animal experiments related with tastes were unusually difficult.
I added these sentences as shown below.
while the mechanism this abnormality of tastes induced by CKD remained unclear
Line 115: I would consider changing the word “sensitivity” to “sensation.”
Ansewer
Thank you for your advice. I changed “sensitivity” to “sensation.” in four places as shown in red words.
Line 129: I think the work “excess” may be appropriate in front of the word sodium.
Ansewer
Thank you for your useful comment. The “excess” made the sentences more clear. I added “excess” in five places as shown in red words/
Line 136: The article that is discussed in this section has both sodium and potassium intake discussed. Is it possible to include a graph that includes both sodium and potassium intake?
Ansewer
Thank you for your useful comment. A graph that includes both sodium and potassium intake is interesting and important, but the previous two reports which I used for making new figures except general population did not include the data concerning potassium intake,
Line 189: the expression “several sodium intakes less than…” wording is confusing. Please consider revising.
Ansewer
Thank you for your useful comment. This part was confusing as you pointed. I changed this part as shown below
However, CKD patients with several categories of sodium intakes less than 194.6 mmol sodium/day (11.4 g salt/day) had almost the same kidney prognosis. Moreover, those CKD patients had the same mortality (see Figure 5).
Line 447: “more than 4 g sodium intake=10 g sodium intake”. Is the equal supposed to be something else?
Ansewer
Thank you for your careful comment. I exchanged ” might be more than 4 g sodium intake=10 g salt intake” to “who might ingest more than 4g sodium a day (=10 g salt intake a day)”
REVIEW2
I would like to thank to the editor the opportunity of reviewing this interesting study. The manuscript is original due to its theme and of great interest to the readers of the journal Nutrients. Overall it seems that a lot of work has been done already. However, authors must clarify/add information about some issues what I indicate below:
- Introduction: The introduction is well written and contextualize adequately the study.
- Method: It is very important that all the methodology described in the study appears.
- The conclusions are very well presented.
Thank you very much for your kind comments.
Reviewer 2 Report
I would like to thank to the editor the opportunity of reviewing this interesting study. The manuscript is original due to its theme and of great interest to the readers of the journal Nutrients. Overall it seems that a lot of work has been done already. However, authors must clarify/add information about some issues what I indicate below:
- Introduction: The introduction is well written and contextualize adequately the study.
- Method: It is very important that all the methodology described in the study appears.
- The conclusions are very well presented.
Author Response

(The authors gave the same response as above.)
